# The magnitude of neonatal near miss and associated factors among live births in public hospitals of Jimma Zone, Southwest Ethiopia, 2020: A facility-based cross-sectional study

**Merertu Wondimu**[1], **Fikadu Balcha**[1], **Girma Bacha**[1], **Aklilu Habte**[2]*

**1** School of Nursing and Midwifery, Faculty of Health Science, Institute of Health, Jimma University, Jimma, Southwest Ethiopia, **2** Department of Public Health, College of Medicine and Health Sciences, Wachemo University, Hosanna, Southern Ethiopia

* akliluhabte57@gmail.com

## Abstract

### Background

Neonates with severe complications at birth or during the neonatal period who nearly died but survived constitute neonatal near miss (NNM) cases. Identifying NNM cases and correcting contributing factors are of the utmost importance to get relevant controls for neonatal deaths. However, limited studies are assessing the prevalence of NNM and associated factors with NNM cases in Ethiopia. So, this study is aimed at assessing the magnitude of neonatal near miss and associated factors among live births in public hospitals of Jimma zone, southwest Ethiopia, 2020.

### Methods

A facility-based cross-sectional study was conducted among 260 neonates from April 1–30 / 2020. Face to face interviewer-administered structured questionnaire was used to collect data from the mothers and a standard checklist was used for their neonates. The data was encoded and entered into Epi-Data version 4.2 and exported to SPSS version 23 for analysis. Independent variables with marginal associations (p-value <0.25) in the bivariable analysis were eligible for multivariable logistic regression analysis to detect an association with outcome variables. Finally, adjusted odds ratios (AOR) with 95% CI were used to estimate the strength of associations, and statistical significance was declared at a p-value < 0.05.

### Result

The magnitude of NNM was 26.7% with [95%CI: 21.6–32.5]. Hypertension during pregnancy [AOR: 3.4; 95%CI: 1.32–8.88], mode of delivery [AOR: 3.32; 95%CI: 1.48–7.45], Obstructed labor [AOR: 2.95; 95%CI: 1.32–6.45] and non-vertex fetal presentation during delivery [AOR: 4.61; 95%CI: 2.16–9.84] were identified as significantly predictors of NNM.

**Data Availability Statement:** All relevant data are within the paper and its Supporting Information files.

**Funding:** The author(s) received no specific funding for this work.

**Competing interests:** The authors have declared that no competing interests exist.

**Abbreviations:** AOR, Adjusted Odds Ratio; ANC, Antenatal Care; APH, Ante-Partum Hemorrhage; C/S, Cesarean Section; CLAP, Latin American Center of Perinatology; CPAP, Continuous Positive Airway Pressure; COR, Crude Odd Ratio; NNM, Neonatal Near Miss; NICU, Neonatal Intensive Care Unit; PLTC, Potentially Life-Threatening Condition; PROM, Premature Rupture of Membrane; WHO, World Health Organization.

## Conclusion and recommendation

Over a quarter of the neonates were with NNM cases, which is relatively higher than the report of studies done in other countries. Hypertension during pregnancy, cesarean delivery, prolonged labor, and non-vertex fetal presentation were all found to increase the likelihood of NNM. Therefore, concerted efforts are needed from local health planners and health care providers to improve maternal health care services especially in early identification of the complications and taking appropriate management.

## Background

The neonatal period is the period from birth to 28 days of life. It is within this period that infants are highly vulnerable to death [1]. Of newborn deaths in the first month of life, about a third of all neonatal deaths tend to occur on the day of birth and close to three quarters die in the first week of life [2, 3]. Preterm birth complications, intrapartum-related complications, sepsis, congenital abnormalities, pneumonia, tetanus, and diarrhea were identified as the major causes of death in neonates [4].

Neonatal near miss (NNM) is a concept related to neonatal mortality where neonates survive either by chance or by the quality of care provided [5]. There is no standard definition or internationally agreed identification criteria for NNM cases, due to this it has been used inconsistently. Some kinds of literature defined it as a newborn who presented with a severe complication/s that occurred during pregnancy, birth, or within 28 days of extra-uterine life but survived [6, 7]. While others are defined as a newborn who nearly died but survived having overcome serious complications during pregnancy, delivery, or within the first seven days of life [8, 9]. However, the Latin American Centre for Perinatology (CLAP) from Pan American Health Organization prepared a standardized definition after reviewing different studies done on NNM as any newborn infant who exhibited pragmatic and/or management criteria and survived the first 27 days of life [10].

Globally 2.5 million neonatal deaths occur with an estimated neonatal mortality rate of 18 deaths per 1,000 live births [4]. Less than 1% of these deaths occur in developed countries [11]. While 98% of all neonatal deaths occur in developing countries, mostly at home, outside the formal health care system. Largely the deaths were related to infections, birth asphyxia and injuries, and consequences of prematurity, low birth weight, and congenital anomalies [12]. In Sub-Saharan Africa (SSA) neonatal mortality rate was 28 deaths per 1,000 live births and a child born in this region has 10 times more likely to die in the first month than a child born in a high-income country [4]. Of neonatal deaths at SSA 50% occurred in just five countries Ethiopia, Nigeria, DR Congo, Tanzania, and Uganda [11]. In addition to this, Ethiopia Mini Demographic Health Survey reported that the neonatal mortality rate remained increasing from 29/1000 LB to 30/1000 LB within the last four years (2015–2019) [13].

Irrespective of decreased neonatal mortality rate both in the developed and developing countries, the neonatal morbidity rate remains elevated [14]. A cross-sectional study done in Brazil showed the estimated number of survivors from NNM cases was seven times higher than the number of neonatal deaths, meaning, that for every neonatal death seven neonates were nearly died but survived [15]. Similarly, in Uganda, the NNM rate was two times higher than that of neonatal mortality rate [16]. Neonates who undergo severe complications might also develop long-term morbidity through effects on neurological and cognitive development and also has associations with chronic diseases such as diabetes, cardiovascular disease, and

chronic lung disease as well as other major disabilities such as blindness or low vision and hearing loss in their late-life [12, 17].

The magnitude of NNM was widely varied across studies because of the difference in criteria used. As studies that used only pragmatic criteria, the incidence of NNM varied between 21.4/1000 live births in Brazil [18] and 86.7/1000 live births in India [19]. Whereas, according to those studies that used both pragmatic and management criteria, the incidence of NNM ranged between 39.2/1000 live births [20] to 367/1000 live births [16].

The UN Agenda for Sustainable Development Goal (SDG) from 2016 to 2030 was to end preventable deaths of newborns and indicated that the neonatal mortality should be less than 12/1000 LB at the end of 2030 [21]. So identifying NNM cases and correcting contributing factors were of the utmost importance to get relevant controls for neonatal deaths, since many babies who die pass through a phase of organ dysfunction before dying and also to prevent long term consequences of severe neonatal morbidity [22].

Investigating NNM cases would aid in taking measures for further amendment of service delivery and programs. This research was therefore intended to determining the prevalence of NNM and to recognize associated factors since the prevalence of NNM is not well understood in Ethiopia. It also helps in recognizing the contributory factors of neonatal mortality and morbidity so that appropriate actions can be adopted at the community and health systems level.

## Methods and materials

### Study design and setting

A facility-based cross-sectional study design was conducted from April 1-30/2020 in Jimma zone, southwest Ethiopia. The zone has a surface area of 119,316 square kilometers. It has 18 woredas and 1 town administration with a total of 555 kebeles (*Kebele is the smallest administrative unit in Ethiopia*) of which 515 of them were rural and 40 were urban. The population projection of 2014/15 of the zone was 2,486,155. The zone has 3 general hospitals, 4 district hospitals, and 1 referral and teaching hospital.

### The population of the study

All live birth neonates delivered at Jimma zone public hospitals were the source population for this study. The study population is comprised of alive neonates in selected hospitals that meet the eligibility criteria. Those mothers who gave birth at home and were critically ill during the data collection time were excluded from the study. Besides, mothers who had twins were also excluded.

### Sample size determination

The study's sample size was determined using a single population proportion formula. Initially, a sample size of 423 was calculated using the following parameters: a 50% prevalence of NNM (because no similar research had been conducted in Ethiopia), a 95% confidence level, 5% margins of error, and a 10% non-response rate. Since the total population is less than 10,000 (N = 534), we used the finite population correction formula. The total population of N = 534 was obtained by averaging the client flow trends for each of the selected hospitals in April and May over the previous three years.

$$\text{nf} = \frac{\text{ni}}{1 + ni/N} = \frac{423}{1 + 423/534} = 236$$

The overall sample size for the study was 260 after accounting for a 10% non-response rate.

## Sampling technique

Of eight governmental hospitals found in the Jimma zone, four hospitals were selected randomly by lottery method. The sample size for each hospital was proportionally allocated by averaging the trends of the previous three years' client flow for each of the selected hospitals in April and May. Neonates were included consecutively at discharge from the postnatal ward and NICU until the determined sample size was reached (S1 Fig).

## Data collection techniques, tools, and personnel

The data from mothers was collected by using a pre-tested, interviewer-administered structured questionnaire which was adapted from relevant literature [9, 10, 22–26]. The tool has generally three parts involving maternal socio-demographic characteristics, reproductive and obstetric history, and medical history during pregnancy. As data collectors, four midwife nurses who have obstetric care experience (one per hospital) and who can speak the local language were recruited. Data was collected through face-to-face interviews with neonates' mothers after the neonates were assured to be survived or at discharge and Maternal charts were reviewed for clarity of diagnosis. As supervisors, two public health professionals who have a bachelor's degree have been recruited.

Near misses' events were identified by data collectors from neonates' medical records according to the criteria of CLAP [10].

## Data quality management

Before the start of data collection, training was offered to data collectors for one day on the purpose of data collection, data collection techniques, the content of the questionnaires, and how to approach the respondents. The data collection tool for the maternal side was prepared in English and translated to the local language Afaan Oromo. Then re-translated back to English to verify the consistency. The pretest was done at Bedele general hospital by taking 13 (5%) of the total sample size before the actual data collection to assess instrument simplicity, flow, and consistency. A day-to-day follow-up during the data collection period was carried out by the principal investigator and supervisors. Every day the collected data was reviewed and cross-checked for completeness and relevance before data entry.

## Data analysis

The data was coded and entered into Epi-Data version 4.2 and exported to statistical package for social science (SPSS) version 23 for analysis. Inconsistencies and missing values were checked by running frequencies. Descriptive statistics like frequency distributions, mean, and standard deviation were computed. The bivariable analysis was done primarily to check the association of each explanatory variable with the outcome variable (NNM). Explanatory variables with marginal associations (p-value <0.25) in the bivariable analysis were eligible for multivariable logistic regression analysis to identify significant predictors of NNM. Finally, adjusted odds ratios (AOR) with 95% CI were estimated to assess the strength and the direction of associations, and statistical significance was declared at a p-value < 0.05.

## Variables of the study

**Neonatal near miss.** NNM was considered when the newborn faced at least one of the following proposed criteria (either pragmatic or management criteria) but survived. From pragmatic criteria: Birth weight < 1750g, gestational age < 33 weeks, 5th-minute Apgar score < 7 or from management criteria: parenteral therapeutic antibiotics up to 7 days and before 28

days of life; nasal continuous positive airway pressure; any intubation during the first 28 days of life; phototherapy within the first 24 hours of life; cardiopulmonary resuscitation; the use of vasoactive drugs, anticonvulsants, surfactants, blood products and steroids for refractory hypoglycemia, parenteral nutrition, any surgical procedure, Congenital malformation if considered as a near miss in other criteria's [10].

**Preterm birth.** Birth at a gestational age of 28 weeks to less than 37 weeks.

**Low birth weight.** Defined as a birth weight of a live-born infant less than 2500g irrespective of gestational age.

**Stillbirth.** Defined as the birth of an infant that has died in the womb or during intrapartum after 28 weeks of gestation.

**Apgar score.** Score ranging from 0–10 based on a newborn's tone, color, respiration, pulse rate, and responsiveness at 1, 5, and 10 minutes.

**Birth interval.** The duration between the current birth and the preceding birth.

**Pre-eclampsia.** Persistent systolic blood pressure of 160 mmHg or more or a diastolic blood pressure of 110 mmHg; and either proteinuria of 5 g or more in 24 hours; or oliguria of <400 ml in 24 hours; or HELLP syndrome or pulmonary edema without seizure of eclampsia and/or diagnosed as severe pre-eclampsia case by a physician.

**Eclampsia.** Generalized fits in a patient without previous history of epilepsy includes coma in pre-eclampsia and other causes of seizure were ruled out by a physician.

## Ethical clearance and consent to participate

Ethical clearance was obtained from the Institutional Review Board of Jimma University, Institute of Health. Permission letter was taken from the department of nursing and midwifery and given to selected hospitals. For those aged 18 and over, written informed consent was obtained from study participants. Besides, after explaining the study goals and procedures, written informed consent was taken from a parent or guardian using normal disclosure processes for those participants less than 18 years of age. A specific ID number was allocated to preserve the anonymity of the questionnaire. The privacy and confidentiality of participants were guaranteed before data collection.

## Results

### Socio-demographic characteristics of the respondents

Of a total of 260 sampled respondents, 255 took part in the study and yielded a response rate of 98.1%. The majority of respondents (86.7%) belong to the 20–34 age group, with the mean (±SD) age of (25.5 ± 4.7) years. Almost all (99.6%) of the mothers were married. About four out of ten (40.4%) of mothers were Muslim by religion and more than half (52.5%) were Oromo in ethnicity. Nearly two-thirds (62.4%) were rural residents and one-third (33.3%) of them attended primary education (Table 1).

### Obstetrics characteristics of the respondents

The majority of respondents (65.5%) were multiparous, and nearly two-thirds (63.5%) of mothers had at least one ANC visit during their most recent pregnancy. More than one-third (37.6%) of mothers gave birth with an interval of 24–48 months between the preceding and current birth and 194 (76.1%) of mothers gave birth through spontaneous vaginal delivery. Twenty- five (9.8%) and 10 (3.9%) of the respondents had experienced abortion and stillbirth, respectively (Table 2).

**Table 1. Socio-demographic characteristic of respondents in public hospitals of Jimma zone, southwest Ethiopia, 2020.**

| Variable Categories | Frequency | Percent |
|---|---|---|
| **Age in years (n = 255)** | | |
| 15–19 | 21 | 8.2 |
| 20–34 | 221 | 86.7 |
| 35–49 | 13 | 5.1 |
| **Marital status (n = 255)** | | |
| Married | 254 | 99.6 |
| Single | 1 | 0.4 |
| **Ethnicity (n = 255)** | | |
| Oromo | 134 | 52.5 |
| Amhara | 59 | 23.1 |
| Dawuro | 26 | 10.2 |
| Gurage | 25 | 9.8 |
| Others* | 11 | 4.3 |
| **Religion (n = 255)** | | |
| Muslim | 103 | 40.4 |
| Orthodox | 90 | 35.3 |
| Protestant | 61 | 23.9 |
| Catholic | 1 | 0.4 |
| **Maternal educational level (n = 255)** | | |
| No formal education | 56 | 22 |
| Can read and write only | 36 | 14.1 |
| Primary (1–8) | 85 | 33.3 |
| Secondary (9–12) | 47 | 18.4 |
| College and above | 31 | 12.2 |
| **Paternal educational level (n = 254)** | | |
| No formal education | 26 | 10.2 |
| Read and write only | 30 | 11.8 |
| Primary (1–8) | 72 | 28.3 |
| Secondary (9–12) | 74 | 29.1 |
| College and above | 52 | 20.5 |
| **Mother's occupation (n = 255)** | | |
| Housewife | 174 | 68.2 |
| Merchant | 41 | 15.3 |
| Government employee | 28 | 11 |
| Others ** | 12 | 4.6 |
| **Paternal occupation (n = 254)** | | |
| Farmer | 70 | 27.6 |
| Merchant | 94 | 37 |
| Government employee | 47 | 18.5 |
| Daily laborer | 6 | 2.4 |
| Private employee | 25 | 9.8 |
| Other *** | 12 | 4.7 |
| **Residence (n = 255)** | | |
| Urban | 96 | 37.6 |
| Rural | 159 | 62.4 |
| **Average monthly income (n = 255)** | | |

*(Continued)*

**Table 1.** (Continued)

| Variable Categories | Frequency | Percent |
|---|---|---|
| < = 2000 | 42 | 16.5 |
| 2001–3500 | 89 | 34.9 |
| 3501–5000 | 63 | 24.7 |
| = >5001 | 61 | 23.9 |

* Tigrai, Yem, Kaffa

** students, daily laborer

***Unemployed, NGO

## Newborn related characteristics

Of 255 selected neonates 152 (59.6%) of them were females and 216 (84.7%) of the neonates' presentation during delivery was vertex.

## The magnitude of neonatal near miss (NNM) conditions

The magnitude of neonatal near miss (NNM) in the study area was 26.7% (95%CI: 21.6%-32.5%). Of the management criteria, cardiopulmonary resuscitation (CPR) was the commonest service obtained by 22 (8.6%) of neonates with near miss conditions closely followed by the use of anticonvulsant 21 (8.2%). From pragmatic criteria, an APGAR score of less than 7 was the most common near-miss condition sustained by 13 (5.1%) of neonates. Unidentified criteria were the use of surfactants and vasoactive drugs (Table 3).

## Factors associated with NNM

In bivariable logistic regression analysis, ten variables namely; maternal age, mother's level of education, father's level of education, mode of delivery, obstructed labor, prolonged labor, hypertension, having urinary tract infection during pregnancy, fetal presentation at delivery, and sex of the newborn had shown association at p-value <0.25 and were a candidate for the multivariable logistic regression model. In multivariable logistic regression analysis hypertension during pregnancy, mode of delivery, prolonged labor, and non-vertex fetal presentation during delivery were identified as significant predictors of NNM.

For those mothers with hypertension during pregnancy, the odds of having NNM was 3.4 times higher than their counterparts [AOR: 3.4; 95%CI: 1.32–8.88]. Being an NNM has been significantly associated with obstetric complications like obstructed labor during the last delivery. In contrast to those women with normal labor, the likelihood of NNM was just about 3 times higher among women with obstructed labor [AOR: 2.95; 95%CI: 1.32–6.45]. As a factor influencing the occurrence of the NNM condition, a fetal presentation was also identified. Compared to those with vertex presentation, neonates that had a non-vertex presentation were 4.6 times more likely to sustain a near-miss event [AOR: 4.61; 95%CI: 2.16–9.84]. On the other hand, in contrast to those mothers who gave birth through spontaneous vaginal delivery, the probability of having an NNM was 3.3 times higher among those mothers who gave birth by cesarean section [AOR: 3.3; 95% CI: 1.48–7.45] (Table 4).

## Discussion

This study was conducted to determine the magnitude of NNM and associated factors at selected public hospitals in Jimma zone, southwest Ethiopia. The finding of this study shows that the magnitude of NNM was 26.7% with 95% CI: (21.6%-32.5%). This finding is

**Table 2. Obstetric characteristics of respondents in selected public hospitals of Jimma zone, southwest Ethiopia, 2020.**

| Variable Categories (n = 255) | Frequency | Percent |
|---|---|---|
| **Gravidity** | | |
| Primigravida | 88 | 34.5 |
| Multigravida > = 2 | 167 | 65.5 |
| **Parity** | | |
| Primipara | 90 | 34.3 |
| Multipara 2–4 | 150 | 58.8 |
| Grand multipara > = 5 | 15 | 5.9 |
| **History of stillbirth** | | |
| Yes | 10 | 3.9 |
| No | 245 | 96.1 |
| **History of abortion** | | |
| Yes | 25 | 9.8 |
| No | 230 | 90.2 |
| **History of preterm birth** | | |
| Yes | 6 | 2.4 |
| No | 249 | 97.6 |
| **History of neonatal death** | | |
| Yes | 10 | 3.9 |
| No | 245 | 96.1 |
| **Birth interval** | | |
| <24 | 48 | 18.8 |
| 24–48 | 96 | 37.6 |
| >48 | 21 | 8.2 |
| **Frequency of ANC follow up** | | |
| No ANC visit | 9 | 3.5 |
| 1–3 times | 162 | 63.5 |
| 4 and above | 84 | 33.0 |
| **Mode of delivery** | | |
| SVD | 194 | 76.1 |
| Instrumental delivery | 24 | 9.4 |
| Cesarean delivery | 37 | 14.5 |
| **Prolonged labor** | | |
| Yes | 36 | 14.1 |
| No | 219 | 85.9 |
| **Obstructed labor** | | |
| Yes | 28 | 11 |
| No | 227 | 89 |
| **Hypertension during pregnancy** | | |
| Yes | 22 | 8.6 |
| No | 233 | 91.4 |
| **Urinary tract infection** | | |
| Yes | 235 | 7.8 |
| No | 20 | 92.2 |
| **Premature rupture of membrane (PROM)** | | |
| Yes | 9 | 3.5 |
| No | 246 | 96.5 |

(*Continued*)

**Table 2.** (Continued)

| Variable Categories (n = 255) | Frequency | Percent |
|---|---|---|
| **Antepartum hemorrhage (APH)** | | |
| Yes | 6 | 2.4 |
| No | 249 | 97.6 |

comparable with the finding of the studies that were conducted in Brazilian university hospitals 30.37% [7] and northeastern Brazil 22% [6].

However, the current finding is lower when compared to the study done in Uganda which was 36.7% [16]. This might be due to a study carried out in Uganda was among mothers with serious obstetric complications, and these complications during pregnancy, labor, and delivery could lead to life-threatening conditions in neonates and place them in the NNM.

On the other hand, the prevalence of NNM in the current study was greater than the finding of the studies conducted in Brazil that reported prevalence of NNM from 3.3% to 8.6% [18, 24, 26, 28] and in India 8.76% [27]. These differences might be due to differences in socio-economic characteristics of the study population, health care delivery system (technologies, early detection of problems). Furthermore, the discrepancy may also be attributable to the criteria used in the identification of NNM cases in which studies conducted in the southeast and northeast Brazil and India used only pragmatic criteria to identify NNM [18, 27, 28] (whereas both pragmatic and management criteria were used in the current study.

Hypertension increased the odds of NNM by three times as compared to those mothers who had no history of hypertension during pregnancy. This result is in line with the finding of studies done in Brazil [25, 28]. This might be due to hypertension during pregnancy may cause complications to fetuses during intrauterine life like intrauterine growth restriction and in extrauterine life such as preterm delivery which is more likely to be LBW and also causes birth asphyxia [29].

In this study, non-vertex fetal presentation during delivery had found to increase the chance of developing NNM. Similarly, the study conducted in Gamo Gofa, Ethiopia, found non-vertex fetal presentation was the determinant factor of NNM [23]. This might be since

**Table 3. Distribution of neonatal near miss conditions among neonates delivered in selected public hospitals of Jimma zone, southwest Ethiopia, 2020 (n = 255).**

| Neonatal near miss (NNM) criteria | Frequency | Percent |
|---|---|---|
| **Pragmatic criteria** | | |
| APGAR score less than 7 | 13 | 5.1 |
| Birth weight less than 1750g | 10 | 3.9 |
| Gestational age less than 33 weeks | 8 | 3.1 |
| **Management criteria** | | |
| Cardiopulmonary resuscitation | 22 | 8.6 |
| Use of anticonvulsant | 21 | 8.2 |
| Use of phototherapy in the first 24 hours | 15 | 5.9 |
| Use of intravenous antibiotic up to 7 days and before 28 days of life | 13 | 5.1 |
| Use of corticosteroid for the treatment of refractory hypoglycemia | 11 | 4.3 |
| Nasal continuous positive airway pressure (NCPAP) | 10 | 3.9 |
| Any surgical procedure | 8 | 3.1 |
| Congenital malformation | 8 | 3.1 |
| Transfusion of blood derivatives | 6 | 2.4 |
| Any intubation | 6 | 2.4 |

**Table 4. Factors associated with NNM in selected public hospitals of Jimma zone, southwest Ethiopia, 2020 (n = 255).**

| Variable Categories | NNM | | COR 95% CI | AOR 95% CI |
|---|---|---|---|---|
| | No (%) | Yes (%) | | |
| **Age** | | | | |
| 15–19 | 13 (61.9) | 8 (38.1) | 1 | 1 |
| 20–34 | 163 (73.8) | 58 (26.2) | 0.578 (0.228–1.466) | 0.779 (0.271–2.238) |
| 35–49 | 11 (84.6) | 2 (15.4) | 0.295 (0.052–1.692) | 0.207 (0.029–1.494) |
| **Mother's educational level** | | | | |
| No formal education | 36 (64.3) | 20 (35.7) | 1 | 1 |
| Can read and write only | 28 (77.8) | 8 (22.2) | 0.514 (0.197–1.339) | 0.565 (0.177–1.802) |
| Primary (1–8) | 64 (75.3) | 21 (24.7) | 0.591 (0.283–1.233) | 0.656 (0.257–1.676) |
| Secondary (9–12) | 34 (72.3) | 13 (27.7) | 0.688 (0.297–1.596) | 0.845 (0.277–2.578) |
| College and above | 25 (80.7) | 6 (19.3) | 0.432 (0.152–1.229) | 0.364 (0.082–1.614) |
| **Paternal educational level** | | | | |
| No formal education | 15 (57.7) | 11 (42.3) | 1 | 1 |
| Can read and write only | 22 (73.3) | 8 (26.7) | 0.496 (0.161–1.524) | 0.484 (0.135–1.706) |
| Primary (1–8) | 51 (70.8) | 21 (29.2) | 0.561 (0.222–1.422) | 0.692 (0.237–1.898) |
| Secondary (9–12) | 59 (79.7) | 15 (20.3) | 0.347 (0.142–1.960) | 0.331 (0.113–1.971) |
| College and above | 40 (76.9) | 12 (23.1) | 0.409 (0.149–1.124) | 0.511 (0.168–1.551) |
| **Mode of delivery** | | | | |
| SVD | 151 (77.8) | 43 (22.2) | 1 | 1 |
| Instrumental delivery | 16 (66.7) | 8 (33.3) | 1.756 (0.704–4.379) | 1.99 (0.74–5.35) |
| Cesarean section | 20 (54.1) | 17 (45.9) | 2.985 (1.439–6.194) | **3.326 (1.485–7.451)**[*] |
| **Obstructed labor** | | | | |
| No | 173 (76.2) | 54 (23.8) | 1 | 1 |
| Yes | 14 (50.0) | 14 (50.0) | 3.204 (1.438–7.139) | 0.630 (0.108–3.683) |
| **Fetal presentation** | | | | |
| Vertex | 169 (78.2) | 47 (21.8) | 1 | 1 |
| Non-vertex | 18 (46.1) | 21 (53.9) | 4.195 (2.067–8.513) | **4.614 (2.163–9.84)**[*] |
| **Prolonged labor** | | | | |
| No | 168 (76.7) | 51 (23.83) | 1 | 1 |
| Yes | 19 (52.8) | 17 (47.2) | 2.947 (1.427–6.089) | **2.959 (1.318–6.595)**[*] |
| **Hypertension during pregnancy** | | | | |
| No | 176 (75.5) | 57 (24.5) | 1 | 1 |
| Yes | 11 (50.0) | 11 (50.0) | 3.088 (1.271–7.500) | **3.421 (1.318–8.881)**[*] |
| **Urinary tract infection (UTI)** | | | | |
| No | 175 (74.5) | 60 (25.5) | 1 | 1 |
| Yes | 12 (60.0) | 8 (40.0) | 1.944 (0.758–4.985) | 1.356 (0.435–4.228) |
| **Sex of the newborn** | | | | |
| Female | 116 (76.3) | 36 (23.7) | 1 | 1 |
| Male | 71 (68.9) | 32 (31.1) | 0.689 (0.393–1.206) | 0.546 (0.290–1.025) |

**Key:** 1: Reference category; AOR = Adjusted odds ratio, COR = Crude odds ratio, *Statically significant at p-value<0.05*

malpresentation during pregnancy and labor has a high risk of birth asphyxia, birth trauma, and other complications and also lead to obstructed and prolonged labor which can result in different complications to the newborn [30].

Obstetric complications during labor and delivery were also showed a significant association with NNM. This study revealed that the odds of NNM was 3 times higher in mothers with obstructed labor when compared to mothers with normal labor. This might be because abnormal

progress of labor that diminishes uteroplacental blood flow can cause fetal distress, fetal hypoxia, and other complications that predispose neonates to life-threatening conditions [31].

In this study cesarean mode of deliveries was associated with an increased risk of NNM. This is in line with the finding of a study done in Southern Ethiopia [23] and studies done in Brazil [15, 20, 25]. Various studies also found that cesarean delivery was associated with an increased risk of APGAR score less than 7 at the 5th minute, preterm birth, low birth weight, neonatal resuscitation, and admission to neonatal intensive care units (NICU), all of which collectively increased the tendency of becoming a near miss [32–34].

The strength of this study was that validated and standardized Neonatal Near Miss identification criteria were used to reduce misclassification and also cross-checked maternal medical records to mitigate recall bias and enhance its validity. The study, however, has some limitations as the neonates were only sampled from hospitals, which may lead to underestimation of the prevalence of NNM as mothers who deliver at home and low-level health facilities (facilities without NICU) were not included in the study because it is difficult to obtain information on the condition of newborns at birth like; APGAR scores at 5th minute, birth weight and gestational age for those neonates delivered at home.

## Conclusion

The magnitude of NNM in the study area was found to be high compared to most studies. In this study hypertension during pregnancy, prolonged labor, cesarean mode of delivery, and non-vertex fetal presentation during delivery were significantly associated with being a near miss. Therefore, concerted efforts are needed from local health planners and health care providers to improve maternal health care services especially in early identification of the complications and taking appropriate management. And also further research is needed to identify other factors by using other study designs.

## Supporting information

**S1 Fig. Schematic representation of the sampling procedure followed to get study participants in Jimma zone, Southwest Ethiopia, 2020.**
(TIF)

**S1 Dataset. The raw data supporting the findings of this article.**
(SAV)

**S1 Questionnaire. Data collection tool for the study.**
(DOCX)

## Acknowledgments

We are indebted to Jimma University Institute of Health for giving the Ethical clearance to undertake the study. Our appreciation also goes to the managers and healthcare providers who worked in the Jimma Zone Health Office for their assistance and cooperation during the study. Finally, we are grateful to our data collectors, supervisors, and the study participants for their efforts throughout the study.

## Author Contributions

**Conceptualization:** Merertu Wondimu, Fikadu Balcha, Girma Bacha, Aklilu Habte.

**Data curation:** Merertu Wondimu.

**Formal analysis:** Merertu Wondimu, Fikadu Balcha, Girma Bacha, Aklilu Habte.

**Investigation:** Merertu Wondimu, Girma Bacha.

**Methodology:** Merertu Wondimu, Fikadu Balcha, Girma Bacha, Aklilu Habte.

**Project administration:** Merertu Wondimu.

**Resources:** Merertu Wondimu.

**Supervision:** Merertu Wondimu, Fikadu Balcha, Girma Bacha, Aklilu Habte.

**Visualization:** Merertu Wondimu.

**Writing – original draft:** Merertu Wondimu, Aklilu Habte.

**Writing – review & editing:** Merertu Wondimu, Fikadu Balcha, Girma Bacha, Aklilu Habte.

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
