## [Decision Letter · Decision Letter 0]

22 Apr 2021

PONE-D-21-03070

The Magnitude of Neonatal Near Miss and Associated Factors among Live Births in Public Hospitals of Jimma Zone, Southwest Ethiopia, 2020: A Facility-based cross-sectional study

PLOS ONE

Dear Dr. Hailegebireal,

Thank you for submitting your manuscript to PLOS ONE. After careful consideration, we feel that it has merit but does not fully meet PLOS ONE’s publication criteria as it currently stands. Therefore, we invite you to submit a revised version of the manuscript that addresses the points raised during the review process.

The statistical concern arisen from our statistical editor. Please clarify these concerns pointed out.

The contents and purpose is well understood.

We look forward to receiving your revised manuscript.

Kind regards,

Kazumichi Fujioka

Academic Editor

PLOS ONE

Journal Requirements:

3. Please include in your Methods section (or in Supplementary Information files) the participating hospitals/ institutions.

Furthermore, please provide additional details on the participant recrtuiemnt, proceeds including the inclusion and exclusion criteria used.

Finally, please clarify how informed ascent was documented for participants under the age of 18 years (ie written or verbal, if verbal please stated how verbal consent was recorded).

Reviewers' comments:

Reviewer's Responses to Questions

**Comments to the Author**

1. Is the manuscript technically sound, and do the data support the conclusions?

Reviewer #1: Yes

Reviewer #2: Yes

Reviewer #3: Partly

2. Has the statistical analysis been performed appropriately and rigorously? 

Reviewer #1: Yes

Reviewer #2: I Don't Know

Reviewer #3: No

3. Have the authors made all data underlying the findings in their manuscript fully available?

Reviewer #1: Yes

Reviewer #2: Yes

Reviewer #3: No

4. Is the manuscript presented in an intelligible fashion and written in standard English?

Reviewer #1: Yes

Reviewer #2: Yes

Reviewer #3: Yes

5. Review Comments to the Author

Reviewer #1: Good start for unsual topic , I would suggest to have a wider study covering bigger population with defferent ethnic groups and socieconomic status , Ante natal care and follow up is crucial in such studay and has to be focused on

Reviewer #2: this article is very valuable because Neonatal Near Miss( NNM) data has not been collected in developing countries. Sample size is very large (260 cases) and very precisely analyzed. Even though NNM is due to only the maternal pregnancy conditions (social factors are very important) , the value of this study won't fade.

Reviewer #3: This manuscript reports the percentage of neonatal near miss and investigates the associated risk factors in public hospitals of Jimma Zone, Southwest Ethiopia with a facility-based cross-sectional study in 2020. I have below comments.

For the sample size, with the given assumptions (the prevalence of NNM of 50%, 5% margins of error), the sample size should have been 264 to guarantee the width of 95% confidence interval within +/-5% (corresponding to the planed 5% margins of error). With 10% non-response rate, the final sample size of 293 should have been planned. Please explain the discrepancy sample size calculation. N=255 to 260 would be good for a prevalence of NNM of ~20%. From the results on page 10, the NNM% in this study area was 26.7%. Is it a post-study calculation for the 5% margins of error?

From the results on page 10, the NNM% in the study area was 26.7% and (95%CI: 21.6%-32.5). Please add % after 32.5.

Are the selected four hospitals comparable on clinical function and the level of health care? Please provide the NNM% by hospital and compare other obstetric characteristics of respondents among hospitals. Are there any potential cluster effect by hospital? If yes, the cluster effect needs to be considered in the logistic regression.

6. PLOS authors have the option to publish the peer review history of their article (what does this mean?). If published, this will include your full peer review and any attached files.

Reviewer #1: **Yes: **Sameh Abozaid

Reviewer #2: No

Reviewer #3: No

---

## [Author Response · Author response to Decision Letter 0]

23 Apr 2021

The response to reviews have been attached as a "Response to Reviewers" in the submission process.

---

## [Decision Letter · Decision Letter 1]

29 Apr 2021

The Magnitude of Neonatal Near Miss and Associated Factors among Live Births in Public Hospitals of Jimma Zone, Southwest Ethiopia, 2020: A Facility-based cross-sectional study

PONE-D-21-03070R1

Dear Dr. Hailegebireal,

We’re pleased to inform you that your manuscript has been judged scientifically suitable for publication and will be formally accepted for publication once it meets all outstanding technical requirements.

Kind regards,

Kazumichi Fujioka

Academic Editor

PLOS ONE

Additional Editor Comments (optional):

Reviewers' comments:

Reviewer's Responses to Questions

**Comments to the Author**

1. If the authors have adequately addressed your comments raised in a previous round of review and you feel that this manuscript is now acceptable for publication, you may indicate that here to bypass the “Comments to the Author” section, enter your conflict of interest statement in the “Confidential to Editor” section, and submit your "Accept" recommendation.

Reviewer #3: All comments have been addressed

2. Is the manuscript technically sound, and do the data support the conclusions?

Reviewer #3: (No Response)

3. Has the statistical analysis been performed appropriately and rigorously? 

Reviewer #3: (No Response)

4. Have the authors made all data underlying the findings in their manuscript fully available?

Reviewer #3: (No Response)

5. Is the manuscript presented in an intelligible fashion and written in standard English?

Reviewer #3: (No Response)

6. Review Comments to the Author

Reviewer #3: (No Response)

7. PLOS authors have the option to publish the peer review history of their article (what does this mean?). If published, this will include your full peer review and any attached files.

Reviewer #3: No

---

## [Editor Report · Acceptance letter]

3 May 2021

PONE-D-21-03070R1 

The Magnitude of Neonatal Near Miss and Associated Factors among Live Births in Public Hospitals of Jimma Zone, Southwest Ethiopia, 2020: A Facility-based cross-sectional study 

Dear Dr. Habte:

I'm pleased to inform you that your manuscript has been deemed suitable for publication in PLOS ONE. Congratulations! Your manuscript is now with our production department. 

Kind regards, 

on behalf of

Dr. Kazumichi Fujioka 

Academic Editor

PLOS ONE